# Differential allelic representation (DAR) identifies candidate eQTLs and improves transcriptome analysis

**Lachlan Baer**[1], **Karissa Barthelson**[1,2], **John H. Postlethwait**[3], **David L. Adelson**[1], **Stephen M. Pederson**[4,5,6‡], **Michael Lardelli**[1‡*]

1 School of Biological Sciences, University of Adelaide, Adelaide, South Australia, Australia, 2 Childhood Dementia Research Group, College of Medicine and Public Health, Flinders Health and Medical Research Institute, Flinders University, Bedford Park, South Australia, Australia, 3 Institute of Neuroscience, University of Oregon, Eugene, Oregon, United States of America, 4 Black Ochre Data Labs, Indigenous Genomics, Telethon Kids Institute, Adelaide, South Australia, Australia, 5 Dame Roma Mitchell Cancer Research Laboratories, Adelaide Medical School, University of Adelaide, Adelaide, South Australia, Australia, 6 John Curtin School of Medical Research, Australian National University, Canberra, Australian Capital Territory, Australia

‡ These authors are joint senior authors on this work.
* michael.lardelli@adelaide.edu.au

**Data Availability Statement:** Availability of code All code to reproduce the described analyses can be found below. • APOE mouse dataset: https://github.com/baerlachlan/211001_APOE_Mm • psen1

## Abstract

In comparisons between mutant and wild-type genotypes, transcriptome analysis can reveal the direct impacts of a mutation, together with the homeostatic responses of the biological system. Recent studies have highlighted that, when the effects of homozygosity for recessive mutations are studied in non-isogenic backgrounds, genes located proximal to the mutation on the same chromosome often appear over-represented among those genes identified as differentially expressed (DE). One hypothesis suggests that DE genes chromosomally linked to a mutation may not reflect functional responses to the mutation but, instead, result from an unequal distribution of expression quantitative trait loci (eQTLs) between sample groups of mutant or wild-type genotypes. This is problematic because eQTL expression differences are difficult to distinguish from genes that are DE due to functional responses to a mutation. Here we show that chromosomally co-located differentially expressed genes (CC-DEGs) are also observed in analyses of dominant mutations in heterozygotes. We define a method and a metric to quantify, in RNA-sequencing data, localised differential allelic representation (DAR) between those sample groups subjected to differential expression analysis. We show how the DAR metric can predict regions prone to eQTL-driven differential expression, and how it can improve functional enrichment analyses through gene exclusion or weighting-based approaches. Advantageously, this improved ability to identify probable eQTLs also reveals examples of CC-DEGs that are likely to be functionally related to a mutant phenotype. This supports a long-standing prediction that selection for advantageous linkage disequilibrium influences chromosome evolution. By comparing the genomes of zebrafish (*Danio rerio*) and medaka (*Oryzias latipes*), a teleost with a conserved ancestral karyotype, we find possible examples of chromosomal aggregation of CC-DEGs during evolution of the zebrafish lineage. Our method for DAR analysis

mutant zebrafish dataset: https://github.com/
baerlachlan/210408_psen1_fADfAI • naglu larval
zebrafish dataset: https://github.com/baerlachlan/
211130_Q96K97del_A603fs • sorl1 dataset:
https://github.com/baerlachlan/210216_sorl1_snv.
Statement Two of the RNA-seq datasets involved in
this analysis have already been described in
previously published analyses. For information
about animal ethics, genome editing, breeding
strategy and RNA-seq data generation for each
dataset please refer to their respective publications.
The psen1 mutant zebrafish dataset is described in
Barthelson et al. J Alzheimers Dis. 2021;82(1):327-
347. doi: 10.3233/JAD-210128 and is available in
the GEO database (GSE164466). The APOE mouse
dataset is described in Sullivan et al. J Biol Chem.
1997 Jul 18;272(29):17972-80. doi: 10.1074/jbc.
272.29.17972 and is available at the AD Knowledge
Portal (accession number syn20808171, https://
adknowledgeportal.synapse.org/). The
nagluA603Efs/A603Efs vs. naglu+/+ 7 dpf sibling
larval RNA-seq dataset is available from the GEO
database (GSE217196). It is comprised of RNA-
seq libraries derived from whole zebrafish 7 dpf
sibling larvae either heterozygous for the EOfAD-
like mutation psenQ96_K97del, homozygous for
the MPSIIIB-like mutation nagluA603fs, or wild-
type.

**Funding:** The author(s) received no specific
funding for this work.

**Competing interests:** The authors declare no
competing interests.

requires only RNA-sequencing data, facilitating its application across new and existing
datasets.

## Author summary

Many human-relevant diseases result from genetic mutations that disrupt cellular func-
tions. We can model these mutations in other organisms (e.g. mouse, zebrafish) and
employ gene expression analysis (transcriptomics) to determine how mutations directly
affect cells and how cells adjust expression of their genes to compensate for these muta-
tions. In our transcriptome analyses of dominant disease-causative mutations in zebrafish,
we identified an interesting phenomenon where a disproportionate number of differen-
tially expressed genes reside on the same chromosome as a mutated gene. Here, we pro-
vide strong evidence supporting that the differential expression of some of these
chromosomally co-located genes is not due to the mutation but is due to differential segre-
gation of gene alleles with innately different expression levels (i.e. expression quantitative
trait loci, eQTLs). We have developed a procedure to measure the likelihood of differential
gene expression being due to an eQTL. This allows us to compensate for the presence of
such eQTLs in bioinformatic analyses. Our procedure, Differential Allelic Representation
(DAR) analysis, revealed evidence for aggregation of genes with related functions on the
same chromosome over evolutionary timescales. DAR analysis allows disentanglement of
eQTLs from mutation-dependent gene expression responses, thereby permitting more
comprehensive investigation of transcriptome data.

## Introduction

Morphologies of chromosomes change over evolutionary timescales due to the processes of
translocation, inversion, duplication, and deletion. These processes change the order of genes
along a chromosome, and so change the interactions between those genes. Changes in chro-
mosome morphology create new regulatory relationships between genes and disrupt previous
ones because genes can share regulatory elements and a regulatory element for one gene may
be in an intron of a different nearby gene. The expression of genes is also influenced by their
location within chromatin regulatory domains existing within and between chromosomes
(reviewed by [1]), which are also disrupted after chromosome rearrangements.

To survive and reproduce, living cells/organisms must be resilient in the face of continuous
environmental and genetic change. Such resiliency has driven the evolution of robust systems
of homeostasis. One aspect of homeostasis is the ability to vary the expression of genes in differ-
ent circumstances to maintain appropriate cellular functions. Transcriptome analysis monitors
variations in gene expression at the RNA level. When transcriptome analysis is used to examine
a biological system before and after a change (environmental or genetic), the differences in tran-
script levels reflect both the direct impacts of the change and homeostatic responses to it.

The alleles of genes positioned in cis on the same chromosome may show genetic linkage
leading to non-random co-assortment into gametes [2]. However, despite linkage, the shuf-
fling of alleles between homologous chromosomes through recombination over successive
generations means that the probability of any two particular alleles of two genes being chro-
mosomally co-located in *cis* in a randomly selected individual should be the product of each
allele's frequency in the population. Occasionally, a particular pair of alleles of two

functionally-related genes can act in concert to increase an individual's likelihood of survival and reproduction. When a chromosomal rearrangement causes these two genes to exist within linkage disequilibrium, the pair of alleles are more likely to be co-inherited by the next generation. This non-random co-inheritance of the two alleles bestows a further fitness advantage and the frequency of this pair in the population increases. Positive selection for the pair of alleles can eventually drive the new chromosomal arrangement to fixation–the chromosomal rearrangement becomes homogeneous across the population. New variants may then arise sporadically within the new arrangement.

Sir Ronald Aylmer Fisher FRS is recognized for his very significant contributions to developing the science of statistics and, in particular, for applying statistical methods to the study of genetics. Fisher's work, '*The genetical theory of natural selection*' published in 1930 [3] is regarded as central to the "modern synthesis" reconciling Darwin's theory of evolution with Mendelian genetics. In discussing the phenomenon of linkage disequilibrium between pairs of alleles across a population he asserted that,

> "...*the presence of pairs of factors in the same chromosome, the selective advantage of each [pair] of which reverses that of the other, will always tend to diminish recombination, and therefore to increase the intensity of linkage in the chromosomes of that species.*"

An expected outcome of this tendency, acting in an evolutionary background of substantial chromosome rearrangement, is that rearrangements establishing linkage between functionally-related genes facilitate selectively advantageous linkage disequilibrium. Therefore, we predict that, over evolutionary timescales:

1. Functionally-related genes may group together to show linkage within chromosomes.

2. The smaller the selective advantage of functionally-related gene pairing, the tighter the genetic linkage between them must be to exert positive selective pressure on chromosome structure over evolutionary time.

3. A gene with multiple, distinct functions (each subject to independent homeostatic mechanisms) will drive accumulation of sets of genes distinct for each function near that gene on the same chromosome.

Considerable evidence exists for the clustering of functionally-related genes on chromosomes in both prokaryotes and eukaryotes [4,5]. Such clustering should be revealed as CC-DEGs in transcriptome data when mutations change the activity of those chromosomally co-located genes and initiate homeostatic responses. However, the reality of functional relationships between CC-DEGs is challenged by the possibility that their observed differential expression results from unequal distribution between experimental comparison groups of alleles with discrete levels of expression (due to differences in allelic transcription rate and/or transcript stability), i.e. "expression quantitative trait loci", eQTLs, as illustrated in Fig 1. Indeed, obvious enrichments of DE genes on the same chromosome as a mutation-of-interest have previously been observed in analyses involving comparisons of homozygous mutant with homozygous wild-type individuals [6,7]. We recently observed such CC-DEGs when comparing transcriptomes from pools of embryos either wild-type or homozygous for a loss-of-function mutation in the zebrafish gene *fmr1* (orthologous to the human *FMR1* gene mutated in Fragile X Syndrome) [8]. However, that analysis also revealed enrichment in homozygous mutant embryos for a particular gene set previously seen as affected in *Fmr1* knock-out mice [8,9], supporting the possibility that some of the CC-DEGs do participate in *fmr1* functions, and highlighting challenges in this aspect of transcriptome analysis.

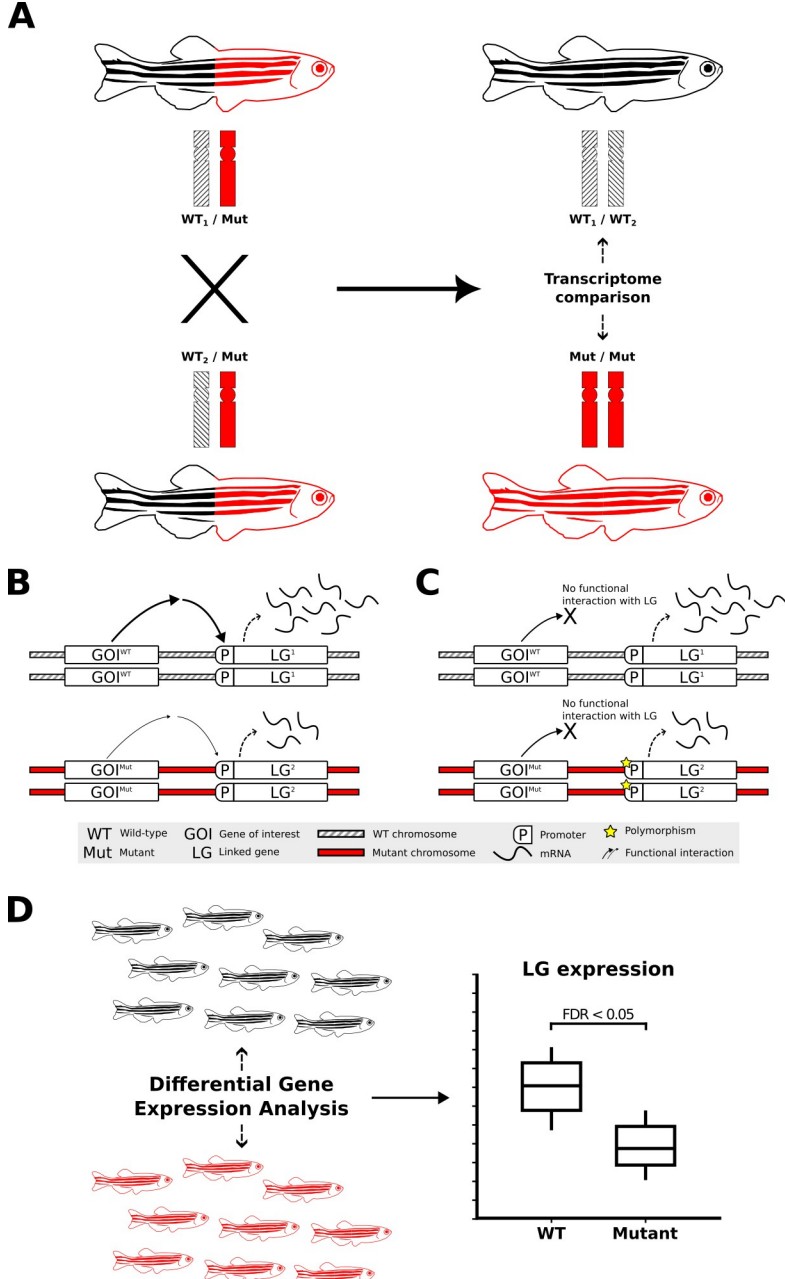

**Fig 1. Transcriptome analysis of homozygous mutants compared to their wild-type siblings and the impact of non-isogenic genetic backgrounds on gene expression. A)** Experimental selection of progeny homozygous for a mutant allele of a gene-of-interest (GOI, mutation-bearing chromosome indicated in red) necessarily involves increased homozygosity for alleles of genes linked to that mutation (i.e. on the same chromosome). The rates of transcription or transcript degradation for these alleles may differ significantly from their corresponding alleles on the homologous wild-type chromosomes (shaded differentially to illustrate that these wild-type chromosomes are not isogenic). **B)** Differential expression of alleles of a linked bystander gene, LG, between wild-type and mutant chromosomes due to a functional interaction between the GOI and the LG. **C)** eQTL-driven differential expression of LG between wild-type and homozygous mutant chromosomes in the absence of a functional interaction between the GOI and the LG. The expression of LG differs independently of the GOI genotype when LG's different alleles are eQTLs. **D)** Breeding to produce genotype groups (e.g., homozygous GOI mutants for comparison to wild-type) selects for differential representation between those groups for the alleles of neighbouring LGs. When those alleles are eQTLs, they can show differential expression between the groups that is not a phenotypic effect of the mutation but can, mistakenly, be inferred as such. Zebrafish icons used in this image were obtained from https://bioicons.com and have been modified from DBCLS https://togotv.dbcls.jp/en/pics.html, licensed under CC-BY 4.0 Unported https://creativecommons.org/licenses/by/4.0/.

White et al. [10] recently published an analysis of gene expression differences between the individual, inbred descendants of an ancestral cross between two discrete isogenic zebrafish strains. This allowed comparison of gene expression from discrete chromosomal segments isogenic for one or other of the ancestral genotypes. The authors demonstrated evidence for the existence and importance of eQTLs in DE gene analysis, and suggested some ways to identify such eQTLs in transcriptome analyses.

Chromosomally co-located allele specific expression has been shown to affect RNA-seq analyses in studies involving recessive mutations [10], but has not yet been investigated for dominant mutations. An early-onset inherited (familial) form of Alzheimer's disease (EOfAD) is caused by dominant alleles of a small number of genes. The majority of EOfAD mutations are missense mutations (or in-frame insertions/deletions) of the human gene *PRESENILIN 1* (*PSEN1*) [11]. In contrast, some frameshift mutations in this gene cause a very distinct, dominant, functional effect—an inflammatory skin disease, familial Acne Inversa (fAI) without Alzheimer's disease [12–14]. We have examined the young adult brain transcriptomes from a number of zebrafish models of these mutations [12,15–22]. As expected from the very different disease phenotypes of human *PSEN1* frame-preserving and frameshift mutations, our previously published analyses of these zebrafish mutation model transcriptomes have revealed very distinct functional effects on energy metabolism and cell signalling pathways [12]. However, among the genes detected as DE between the brains of the wild-type and the dominant heterozygous mutant zebrafish, we were surprised to observe that a greater than expected number were chromosomally colocalised with the mutation-of-interest (i.e. many are CC-DEGs). Here we investigate this colocalization phenomenon further by examining between-group differential allelic representation in brain transcriptome datasets from zebrafish. We also extend our investigation to similar mouse transcriptome datasets. We find strong support for the importance of eQTLs, but also for Fisher's postulate on colocalization of functionally related genes. We show that using differential allelic representation information can improve the detection of significant gene sets.

## Results

### Transcriptome comparisons revealing CC-DEGs

Several datasets were selected to facilitate a comprehensive investigation into the presence of CC-DEGs. A summary can be seen in Table 1.

### CC-DEGs in heterozygous mutant analyses

Our previous observation of CC-DEGs in our analysis of homozygous *fmr1* mutant zebrafish embryos [8] led us to re-examine our numerous adult zebrafish brain transcriptome datasets where we had previously analysed the effects of dominant mutations in heterozygotes. Surprisingly, we saw CC-DEGs in a number of our analyses as illustrated by Manhattan plots of DE gene expression (Figs 2A and S1A–S1C). To confirm that CC-DEGs are not a phenomenon specific to zebrafish, we searched in public databases of transcriptome data for other datasets comparing wild-type and mutant (but non-transgenic) individuals. While suitable datasets were few, we found that an analysis of different *APOE* alleles (human nomenclature to denote the use of humanised alleles) in mice showed CC-DEGs for 3-month-old cerebral cortices homozygous for the Alzheimer's disease-protective ε2 allele (*APOE2*) compared to cortices homozygous for the neutral-risk *ε3* allele (*APOE3*, [23], Fig 2B). Interestingly, CC-DEGs were less obvious in the analysis of cortices homozygous for the Alzheimer's disease-risk *ε4* allele (*APOE4*, S1D Fig), supporting that mutations/alleles that are more disruptive to cell biology cause the appearance of DE genes distributed throughout the genome.

**Table 1. Summary of transcriptome data and experimental comparisons that comprise the analyses reported in the following sections.** Experiments were chosen that utilise zebrafish or mouse as the model organism, and discrete datasets are defined by the mutated gene-of-interest. Differential gene expression testing was performed for each comparison, with the threshold for significance set to a false discovery rate (FDR)-adjusted *p*-value of less than 0.05. The total number of differentially expressed genes (Total DEGs) is listed for each comparison, as well as the number that were chromosomally colocalised with the mutated gene (CC-DEGs). The significance column (Sig.) indicates the Bonferroni-adjusted *p*-values for over-representation of CC-DEGs (Fisher's Exact Test, ***: < 0.001, **: < 0.01, *: < 0.05).

| Organism | Mutant gene | Chromosome | Comparison | Total DEGs | CC-DEGs | Sig. |
|---|---|---|---|---:|---:|---|
| *Danio rerio* | *psen1* | 17 | *T428del/+ vs +/+* | 10 | 4 | *** |
| | | | *W233fs/+ vs +/+* | 11 | 7 | ** |
| | *naglu* | 24 | *A603Efs/A603Efs vs +/+* | 53 | 25 | *** |
| | *sorl1* | 15 | *R122Pfs/+ vs +/+* | 5 | 4 | *** |
| | | | *V1482Afs/+ vs +/+* | 3 | 2 | – |
| *Mus musculus* | *Apoe* | 7 | *APOE2/2 vs APOE3/3 (male)* | 664 | 99 | *** |
| | | | *APOE2/2 vs APOE3/3 (female)* | 1663 | 183 | *** |
| | | | *APOE4/4 vs APOE3/3 (male)* | 3963 | 299 | – |
| | | | *APOE4/4 vs APOE3/3 (female)* | 649 | 49 | – |

White et al.'s analysis [10] of DE genes in the inbred descendants of two wild-type isogenic zebrafish strains strongly supports the conclusion that many CC-DEGs are eQTLs and showed how homozygosity for regions of the genome can enhance observation of allelic expression

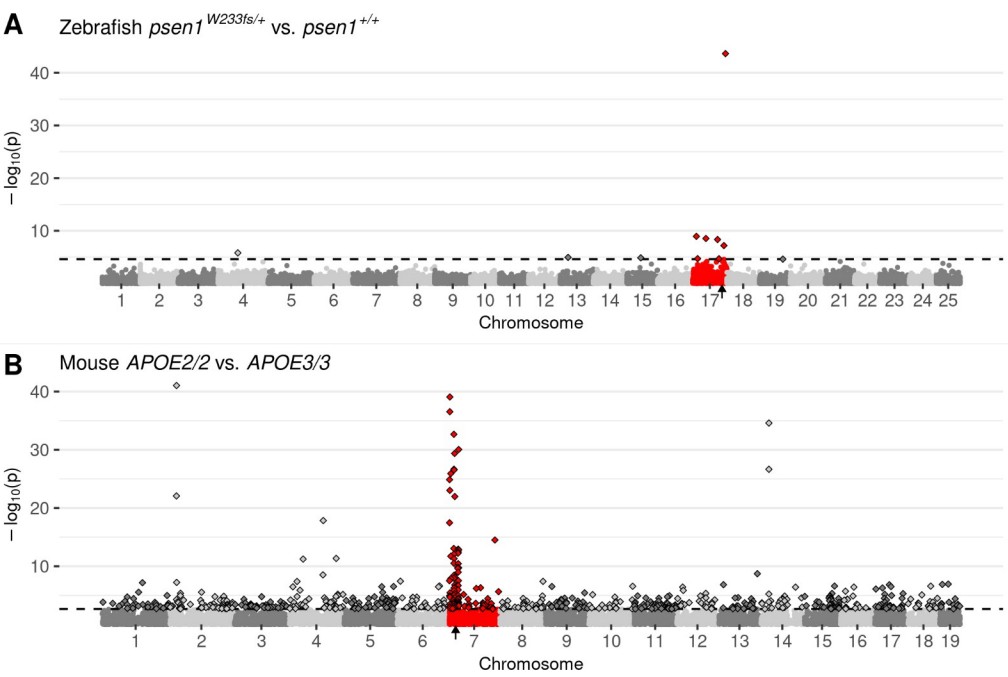

**Fig 2.** Manhattan plots highlighting CC-DEGs from differential expression testing in: **A)** Brain transcriptomes from a comparison of zebrafish *psen1*$^{W233fs/+}$ vs. *psen1*$^{+/+}$ siblings at 6 months of age. *psen1*$^{W233fs}$ is a dominant fAI-like frameshift allele. **B)** 3-month-old cerebral cortex transcriptomes from a comparison of homozygous male mice bearing humanised *APOE2* or *APOE3* alleles. Non-random accumulations of DEGs are supported by Bonferroni-adjusted Fisher's exact test *p*-values for enrichment of DE genes on the mutant chromosome, A) *psen1*$^{W233fs/+}$ vs. *psen1*$^{+/+}$: *p* = 7.41e-7, B) *APOE2/2* vs. *APOE3/3*: *p* = 2.03e-10. Genes plotted along the x-axis based on their genomic positions along the chromosomes distinguished by alternating shades of grey. Genes on the chromosomes containing the mutations are highlighted in red. The approximate locations of the mutated genes are indicated with small black arrows on the x-axis. The raw *p*-values are plotted along the y-axis at the -log$_{10}$ scale such that the most significant genes exist at the top of the plot. The cut-off for gene differential expression (FDR-adjusted *p*-value < 0.05) is indicated by a dashed horizontal line. Genes classified as differentially expressed under this criterion are represented as diamonds with a black outline.

differences. Transcriptome comparisons between individuals or groups of individuals should not reveal eQTLs when comparisons are made within single isogenic (e.g. inbred) backgrounds or when allelic sequence differences between comparison groups (here described as differential allelic representation, DAR) are sufficiently limited. Therefore, we sought to test whether DAR between genotype groups in transcriptome datasets could be used to determine the likelihood that any particular gene is differentially expressed due to eQTL effects. To facilitate this analysis, we assembled a computational workflow to quantify DAR between samples (see below).

## A method for differential allelic representation (DAR) analysis

Differential allelic representation (DAR) occurs when the composition of experimental sample groups results in an uneven distribution of polymorphic loci between these groups. This scenario is commonly encountered in RNA-seq experiments involving organisms that do not share an isogenic background. DAR raises significant concerns regarding the reliability of results in differential expression analysis. In cases where a polymorphic locus involves unequal representations of eQTLs between sample groups, gene expression differences can be observed irrespective of experimental condition, thereby contributing to the set of genes (or transcripts) regarded as differentially expressed. This issue does not pertain to the statistical methods for detecting differential expression, because the underlying biology does indeed result in differential expression of genes. However, the impacts of DAR are confounded with the experimental condition of interest, necessitating the development of a complementary technique alongside differential expression analysis to avoid erroneous inferences.

We developed a computational approach for the examination of DAR in RNA-seq data. DAR analysis yields a metric for each observable polymorphic locus, offering a localised assessment of expression differences that are possibly eQTL-driven across the genome. Notably, our methodology exclusively utilises RNA-seq data, making it applicable to both ongoing and pre-existing RNA-seq investigations.

Our pipeline for calculating DAR between sample groups is summarised in Fig 3 (see Methods for extended details). Briefly, we determined single nucleotide variants based on the GATK Best Practice Workflow for RNA-seq short variant discovery [24]. Monoallelic calls

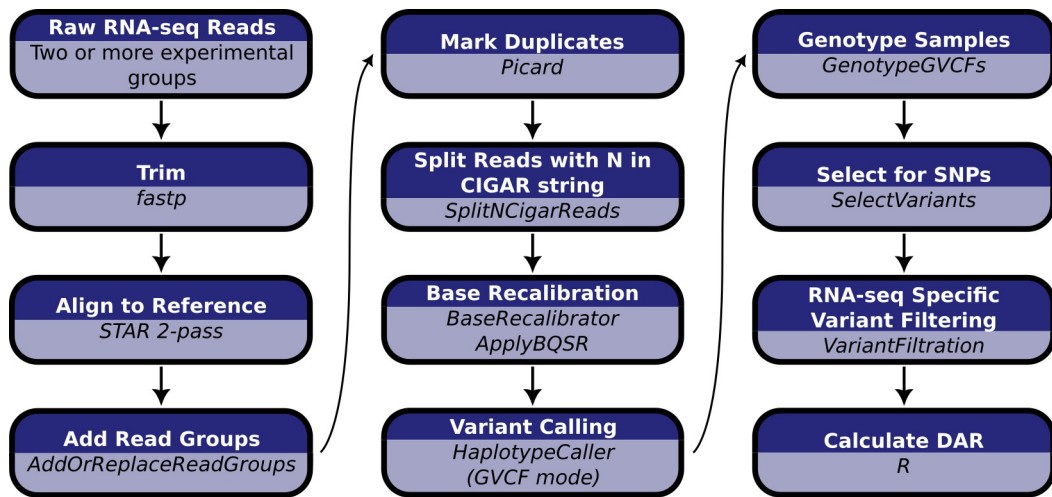

**Fig 3. Computational workflow for the calculation of DAR starting with raw RNA-seq short read data.** Raw RNA-seq reads must consist of at least two experimental groupings to allow the calculation of DAR between them.

(where only the reference allele is reported at a single nucleotide locus across all samples) were excluded due to the computational implications for downstream processing. Bi- and multiallelic calls (one or more alternative alleles reported at a single nucleotide locus across all samples) were then analysed using the R programming language [25]. We present the calculation of the DAR metric as a modification of the Euclidean distance formula,

$$DAR_{1,2} = \sqrt{\frac{\sum_{i \in \{A,C,G,T\}} (p_{i_1} - p_{i_2})^2}{2}}$$

(1)

where $DAR_{1,2}$ represents the DAR value between sample groups 1 and 2 at a single nucleotide locus. $p_{i_1}$ and $p_{i_2}$ correspond to the proportion of the total reported allele counts (A, C, G or T) in sample groups 1 and 2 respectively at a single nucleotide locus. Dividing the Euclidean distance of allele proportions by $\sqrt{2}$ results in an easy-to-interpret DAR metric where 0 represents identical allele proportions and 1 represents complete difference (non-identity).

## DAR analysis of the humanised *APOE* alleles in a mouse cerebral cortex transcriptome dataset

After establishing a workflow to quantify DAR between samples, we quantified variation in this metric across entire genomes with a particular focus on chromosomes containing mutations of interest. We chose the mouse *APOE* cortex transcriptome dataset for an initial DAR analysis as it displayed the strongest signal of CC-DEGs on the mutant chromosome (Fig 2B). For visualisation purposes, and to alleviate noise, we smoothed the DAR metric using an elastic sliding window approach to calculate the mean DAR value at each single nucleotide variant locus along with the nearest five loci on either side (*n* = 11). See S2A Fig for a summary of the elastic window sizes. Interestingly, the cortex transcriptome comparison of mice homozygous for *APOE2* with those homozygous for *APOE3* revealed a region containing peaks of very high DAR localised around the *APOE* gene on Chromosome 7 in both male and female samples (Figs 4 and S3). 99 and 183 CC-DEGs were observed in the comparisons of males and females respectively, with an intersect size of 59 CC-DEGs. An inspection of the cumulative distribution of DAR across each chromosome supports the notion that the mutant chromosome contains the most DAR of all the chromosomes (S4 Fig). In a completely isogenic strain, this result would not be expected. However, the mice in this study were engineered through targeted replacement of the murine *Apoe* locus with human *APOE* alleles by homologous recombination in mouse 129 strain embryonic stem cells followed by repeated backcrossing into the C57BL/6J strain [26]. The strain 129 alleles of genes located near (in tight linkage with) the *APOE* locus are less likely to be replaced by strain C57BL/6J alleles through recombination than alleles of syntenic genes distant from *APOE* or alleles of genes on other chromosomes.

Having identified a region of high DAR potentially prone to eQTL-dependent effects, we looked for CC-DEGs within our differential expression analysis. We observed a clear clustering of CC-DEGs around the *APOE* gene within the identified regions of high DAR, strongly supporting the existence of eQTLs as drivers of differential expression (Fig 4). The remaining DE genes on Chromosome 7 found within regions of low DAR are more likely to represent biological responses to the mutation because they are DE even without close linkage. However, genes may become DE as a secondary effect of eQTLs located elsewhere in the genome [27], so the existence of a DE gene in a region of low DAR does not confirm a direct, primary functional relationship with any mutation. We also noticed that the *APOE* region showed relatively low DAR compared to the surrounding region, which was unexpected as each sample group has a different *APOE* allele. On closer inspection, it was evident that variants in this region

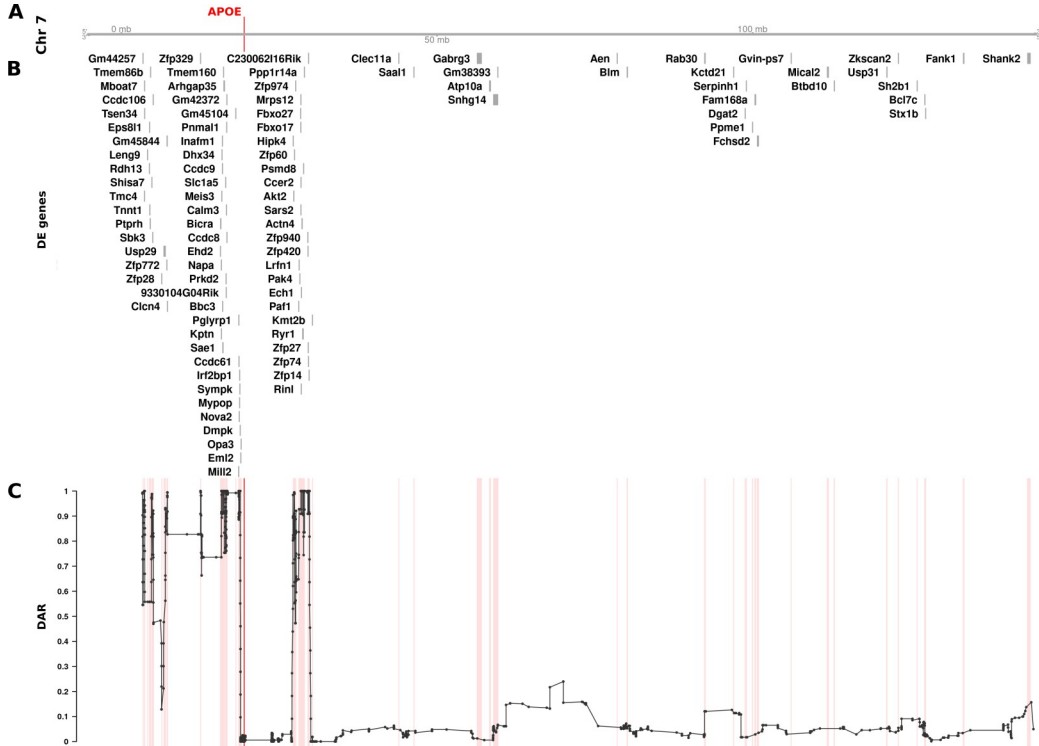

**Fig 4. The relationship between DE genes and DAR along the entirety of mouse Chromosome 7 between male**
***APOE2/2* and *APOE3/3* mouse cerebral cortices (at 3 months of age).** The plot contains four sets of information
represented by separate tracks horizontally. Track A represents the axis of Chromosome 7. The position of the *APOE* gene
is marked and labelled in bold red. Track B displays differentially expressed genes according to their positions along the
chromosome. 99 of 1126 total genes (8.79%) on Chromosome 7 that were expressed in the dataset were classified as DE
(FDR < 0.05). Track C shows the trend in DAR as a connected scatterplot with each point in black representing the DAR
value at a single nucleotide variant position (elastic sliding window size = 11 variants). Positions of the DE genes shown in
track B are indicated by light red lines.

were removed during quality control filtering due to being classified as potential single nucleo-
tide polymorphism (SNP) cluster artefacts (>3 SNPs in a 35bp window). This most likely
occurs because the *APOE* alleles in these mice are humanised and therefore exhibit substantial
variation in the *APOE* region as reads were aligned to the mouse genome (GRCm39, Ensembl
release 104).

## DAR analysis reveals zebrafish *psen1* CC-DEGs are unlikely to be eQTLs

DAR analysis of the mouse *APOE* dataset showed convincing evidence implicating eQTLs as
the primary drivers of differential expression for a large proportion of genes. Unexpectedly,
our analysis of DAR in the comparisons of 6-month-old brain transcriptomes in zebrafish
*psen1*$^{T428del/+}$ (EOfAD-like) vs. *psen1*$^{+/+}$ siblings and in *psen1*$^{W233fsl/+}$ (fAI-like) vs. *psen1*$^{+/+}$
siblings did not support the hypothesis that those DE genes clustered near *psen1* on zebrafish
Chromosome 17 are eQTLs. Nevertheless, the identified DE genes are indeed clustered with
*psen1* on Chromosome 17 (supported by Bonferroni-adjusted Fisher's exact test *p*-values for
enrichment of DE genes on Chromosome 17 despite consistently low DAR along this chromo-
some; *psen1*$^{T428del/+}$ vs. *psen1*$^{+/+}$: *p* = 8.63e-3, *psen1*$^{W233fsl/+}$ vs. *psen1*$^{+/+}$: *p* = 7.41e-7). As shown
in Fig 5, DAR along the length of Chromosome 17 in the *psen1*$^{T428del/+}$ vs. *psen1*$^{+/+}$ and the
*psen1*$^{W233fsl/+}$ vs. *psen1*$^{+/+}$ comparisons was uniformly below 0.2, as expected from the fact that

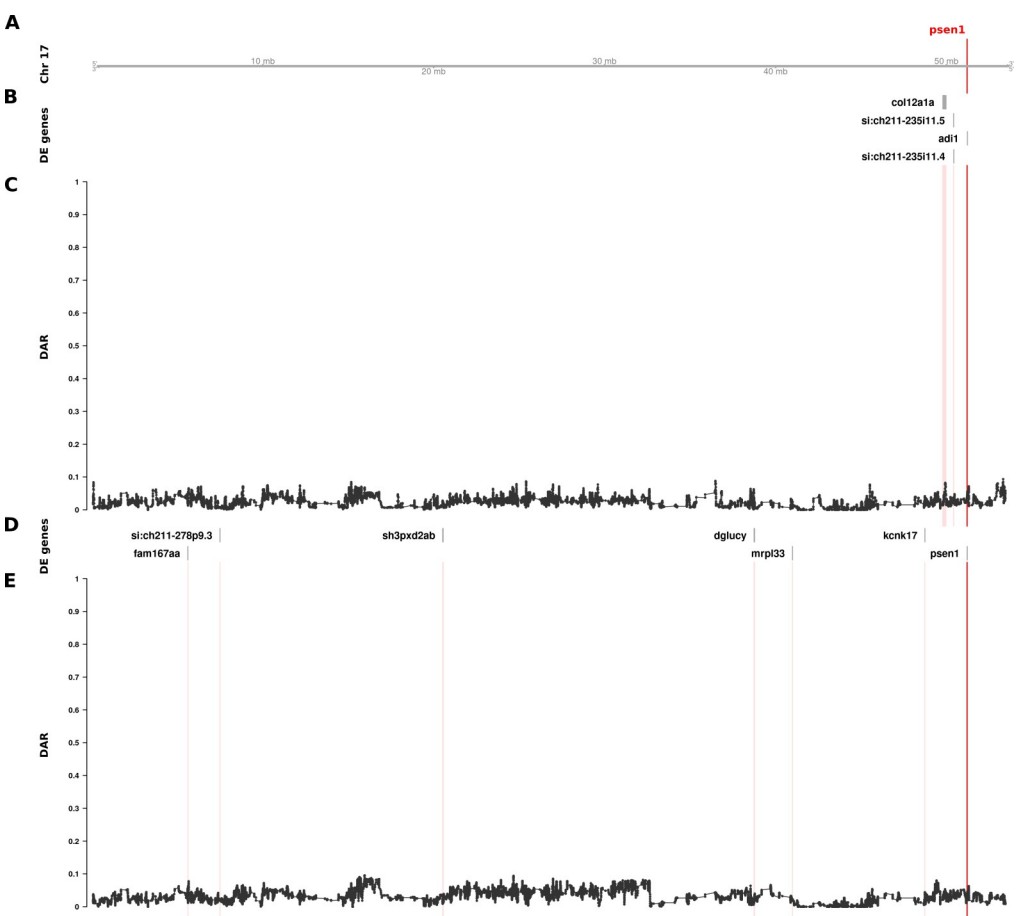

**Fig 5. The relationship between DE genes and DAR along the entirety of zebrafish Chromosome 17 in brain transcriptome comparisons of 6-month-old wild type fish against sibling fish heterozygous for either of two different *psen1* mutations.** Track A represents the axis of Chromosome 17. The position of *psen1* is marked and labelled in bold red. Tracks B and C show the results from the EOfAD-relevant *psen1*$^{T428del/+}$ vs. *psen1*$^{+/+}$ comparison, while tracks D and E show the fAI-relevant *psen1*$^{W233fsl/+}$ vs. *psen1*$^{+/+}$ comparison. Tracks B and D display DE genes from their respective comparisons, while tracks C and E show the trend in DAR as a connected scatterplot with each point representing the DAR value at a single nucleotide variant position (elastic sliding window, n = 11 variants). The positions of DE genes are highlighted on the DAR tracks in a light red colour.

these mutations were generated in one, very inbred, strain of zebrafish. Indeed, the Tübingen (Tu) strain zebrafish in the facility used to produce these mutant fish were bred in relatively small populations in isolation for over 15 years. It is also notable that the genes that were DE due to the EOfAD-like *psen1*$^{T428del}$ mutation were not the same as those DE due to the fAI-like *psen1*$^{W233fs}$ mutation, consistent with these two very different types of mutation in the same gene (reading frame-preserving vs. frameshift respectively) having discrete molecular effects (as revealed and discussed in our previously published analysis [12]). (Note that the mechanisms underlying the differential effects of frame-preserving and frameshift mutations on *PSEN1* function are currently unclear [28]). The genes that were DE due to the EOfAD-like mutation are clustered very close to *psen1*, while those DE due to the fAI-like mutation are greater in number and more widely distributed on Chromosome 17 (Fig 5). This finding is consistent with the second prediction from Fisher's assertion (gene pairings with a smaller selective advantage require tighter genetic linkage, see Introduction) because the relatively late-acting (after reproductive maturity), dementia-inhibiting function of *psen1* that is

disrupted by the EOfAD-like mutation should only exert mild selective pressure on the organism's survival and reproduction. In contrast, the fAI-like mutation affects the role of Psen1 protein in Notch signalling and in other cellular signalling pathways that are important in embryo development and immune system function, and so likely exerts stronger selective effects. It is also relevant to note that meiotic recombination is suppressed in male zebrafish relative to females so that recombination distances along of the majority of chromosomes, including Chromosome 17, are less than 50 centiMorgans in males [29]. This means that, in male zebrafish, alleles of any two genes anywhere on Chromosome 17 are genetically linked.

Due to the low numbers of DE genes in this dataset, we looked for additional evidence that the clustering of DE genes on Chromosome 17 had not occurred by chance. Therefore, we investigated whether DE clustering was still observed after applying a less stringent DE significance threshold. We performed an additional analysis using an FDR-adjusted $p$-value $< 0.15$ (i.e. $3\alpha$ for $\alpha = 0.05$) allowing for detection of additional DE genes, along with a likely increase in false positives. This change increased the total number of DE genes from 10 to 20 across the entire genome in the EOfAD-relevant $psen1^{T428del/+}$ vs. $psen1^{+/+}$ comparison, and from 11 to 30 in the fAI-relevant $psen1^{W233fsl/+}$ vs. $psen1^{+/+}$ comparison. Both comparisons remained significant for enrichment of DE genes on Chromosome 17 with Bonferroni-adjusted Fisher's exact test $p$-values of 1.88e-2 and 2.73e-12 respectively.

## Aggregation of *psen1* CC-DEGs over evolutionary time

Our analysis of zebrafish *psen1* mutant brain transcriptomes supports Fisher's contention that selection for linkage disequilibrium may influence chromosome structure over evolutionary timescales. Therefore, we looked for evidence of structural rearrangements favouring clustering of DE genes during teleost (bony fish) speciation. Medaka (*Oryzias latipes*) is a small freshwater teleost sharing a common ancestor with zebrafish approximately 290 million years ago [30]. Its widespread use as a model organism has encouraged detailed characterisation of its genome. The medaka genome has not undergone any major chromosomal rearrangements in the 323 ± 9.1 million years since divergence from its common ancestor with zebrafish [31]. Therefore, it provides a suitable reference outgroup to assess how the zebrafish genome has changed over time.

We accessed The Synteny Database [32] to assess whether CC-DEGs revealed in our various analyses exist on separate chromosomes in the medaka genome. Although our efforts were limited by low numbers of DE genes, we did indeed find evidence supporting that some *psen1* CC-DEGs have become chromosomally co-located during zebrafish evolution. Of the seven DE genes chromosomally co-located with *psen1* (on zebrafish Chromosome 17) in fAI-like *psen1*^*W233fs*^ mutants, two (*fam167aa*, *sh3pxd2ab*) were predicted to have orthologues on separate medaka primary chromosomes, while another two (*psen1*, *dglucy)* were predicted to be located on alternate scaffolds (S5 Fig). The remaining three DE genes (*mrpl33*, *kcnk17*, *si*: *ch211-278p9.3*) were either not present in the Ensembl version 71 database or not predicted to have medaka orthologues. As Ensembl version 71 was released in 2013 we wanted to confirm our findings with the most recent Ensembl database (version 108, released 2022). By searching the gene names of the zebrafish genes DE on Chromosome 17, we confirmed that *fam167aa* and *sh3pxd2ab* are located on medaka Chromosomes 24 and 1 respectively as predicted by The Synteny Database. However, four DE genes (*psen1*, *dglucy*, *mrpl33*, *kcnk17)* where orthologues could not be found on the medaka primary chromosomes of Ensembl version 71 have since been assigned to medaka Chromosome 22 and represent syntenies since the last common ancestor of medaka and zebrafish. While these four genes on medaka Chromosome 22 have been linked together with *psen1* for at least 323 ± 9.1 million years in zebrafish, *fam167aa*

and *sh3pxd2ab* may be examples of genes that have been captured more recently during zebrafish evolution due to functional interactions with *psen1*.

## DAR-based exclusion of probable eQTLs improves gene set enrichment analyses

eQTL artifacts due to linkage represent noise that could reduce the sensitivity of functional pathway analysis of mutations. Therefore, we investigated whether we could reduce such noise (exclude eQTLs) by excluding genes located in those parts of the genome differing between mutant and wild-type individuals. Using the previously calculated DAR metric across the genome, we set gene exclusion thresholds at DAR values ranging from 0.1 to 1.0 at 0.1 intervals prior to functional enrichment analysis. We excluded from analysis those genes located in regions with a DAR value greater than the threshold. As the maximum possible DAR metric value is 1, a threshold at this value does not remove any genes and, therefore, represents the results of analyses where no eQTLs are excluded. We utilised two well-established and complementary pathway analysis methods at each of the ten thresholds to observe the effects of DAR metric-based filtering; Rotation Gene Set Testing (ROAST) implemented using the *fry* function in Bioconductor package *limma* [33,34], and Gene Set Enrichment Analysis (GSEA) implemented using Bioconductor package *fgsea* [35,36]. We chose the 186 *a priori* gene sets defined by the Kyoto Encyclopedia of Genes and Genomes (KEGG) [37] to be tested in each analysis.

To examine the effects of different DAR metric thresholds, we applied this approach to a zebrafish brain transcriptome dataset previously generated by our laboratory that includes transcriptome data on the effects of homozygosity for a mucopolysaccharidosis type IIIB (MPS IIIB)-like mutation in the gene *naglu*, *naglu*^A603Efs, at 7 days post fertilisation (dpf, see also **Data description and generation**). This dataset displays a highly variable pattern of DAR along the mutant chromosome and clustering of DE genes near the mutation site (Fig 6). Mutations in the human *NAGLU* gene causing MPS IIIB have a clearly understood

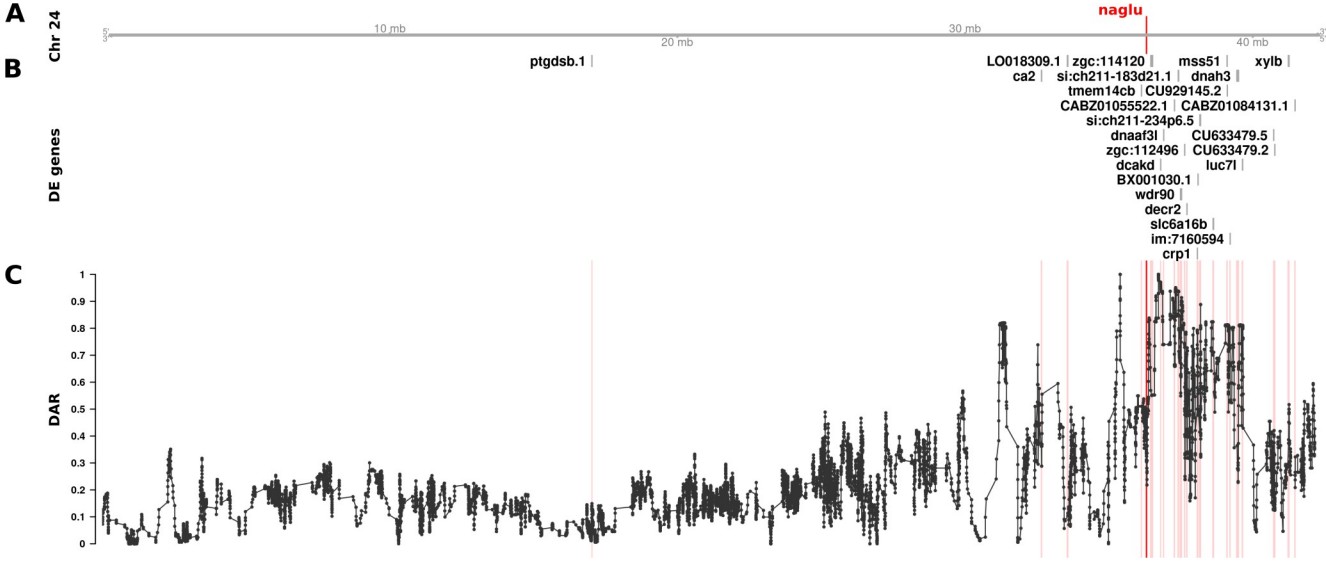

**Fig 6. The relationship between DE genes and DAR along Chromosome 24 between *naglu*^A603Efs/A603Efs and *naglu*^+/+ 7 dpf larval zebrafish.** Track A represents the axis of Chromosome 24. The position of *naglu* is marked and labelled in bold red. Track B displays DE genes according to their positions along the chromosome. Track C shows the trend in DAR as a connected scatterplot with each point representing the DAR value at a single nucleotide variant position (elastic sliding window, n = 11 variants). Positions of the DE genes shown in track B are indicated by light red lines.

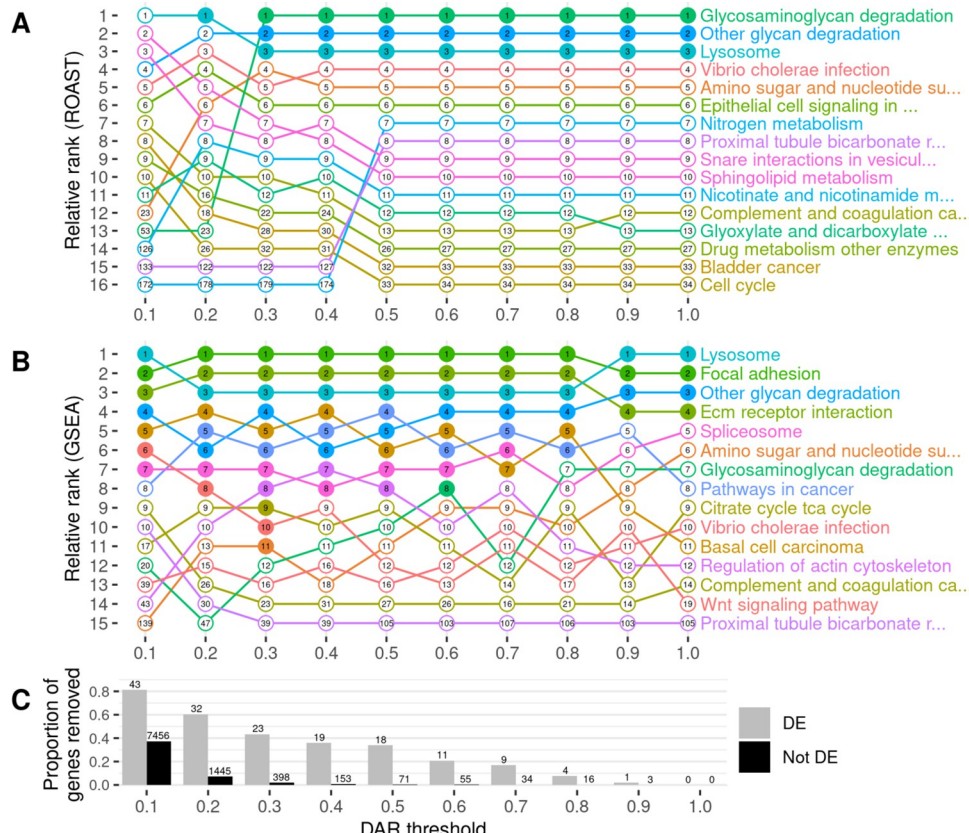

**Fig 7. The impact of gene exclusion by DAR thresholding on the outcomes of functional enrichment analysis using KEGG gene sets.** Panel **A** displays the outcomes of ROAST, while panel **B** displays the outcomes of GSEA. Gene sets are displayed only if they were found in the top ten most significant gene sets for at least one DAR threshold. The relative ranking between the displayed gene sets is represented along the y-axis for each threshold indicated on the x-axis. Filled dots indicate that the gene set was classified as significantly enriched (FDR-adjusted $p$-value < 0.05) at the respective threshold. The numbers inside the dots show the overall ranking of the gene set among all 186 KEGG gene sets tested. Panel **C** displays the proportion of DE (grey) and non-DE (black) genes that were removed at each threshold. The number of genes this equates to is displayed above each bar of the chart.

pathological mechanism; they reduce a lysosomal enzyme activity required in the degradation of heparan sulfate [38]. In contrast to EOfAD, where the pathological mechanism(s) is disputed, interpretation of the results of functional analyses of mutations causing MPS IIIB is less ambiguous. In fact, the KEGG collection of gene sets includes a gene set/pathway directly related to heparan sulfate degradation, "*Glycosaminoglycan Degradation*". We performed functional analysis of *naglu*$^{A603Efs/A603Efs}$ vs. *naglu*$^{+/+}$ siblings from this dataset at each of the aforementioned thresholds and the results are summarised in Fig 7.

Setting DAR thresholds caused only minor impact on the ranking and significance of gene sets when using ROAST. The lower thresholds of 0.1 and 0.2 showed clear perturbations in rankings but this is unsurprising due to the large proportion of genes excluded. Results at low thresholds would be expected to be unreliable due to the removal of genes representing real gene expression differences caused by the mutation. This is evident from the loss of statistical significance of all three pathways that had previously shown significance at thresholds greater than 0.2. The consistent significance of the pathways (gene sets) *Glycosaminoglycan Degradation*, *Other Glycan Degradation*, and *Lysosome* at most thresholds gives confidence in the results of ROAST because these three pathways have previously been implicated in MPS IIIB [39].

In contrast to the effects on ROAST, the imposition of DAR thresholds before GSEA offers a number of interesting observations. Substantial alterations in rankings are seen crossing each threshold for most pathways. This trend remains true even when very few genes are removed from the analysis, for example by DAR thresholds at 0.8 and 0.9 (20 and 4 genes excluded in total respectively). A general trend showing an increasing number of significant gene sets is observed as the DAR metric threshold is lowered. Eleven pathways are classified as significant at a threshold of 0.3 in comparison to four pathways when no filtering was performed (i.e. at a DAR threshold = 1). Most notably, the pathway *Glycosaminoglycan Degradation* that is relevant in MPS IIIB achieves statistical significance uniquely at a DAR threshold of 0.6 where a total of 66 genes (11 DE, 55 not DE) are excluded from the analysis. Interestingly, and despite being an expected true positive, this pathway does not achieve significance at any other threshold. Closer inspection of the *p*-values of pathways that achieved significance at one threshold or more (S6 Fig), found that most pathways return smaller *p*-values as the DAR metric threshold is lowered, until too many genes are excluded. The *Lysosome* and *Regulation of Actin Cytoskeleton* pathways show this trend even as the number of leading-edge genes (those driving the enrichment) become fewer.

Due to our finding that GSEA produced a greater number of significant gene sets at lower DAR thresholds, we examined more closely the genes remaining after imposition of a DAR threshold = 0.3. Interestingly, Fisher's exact test revealed that significant enrichment for clustering of DE genes on Chromosome 24 persisted (Bonferroni-adjusted *p*-value = 6.16e-3) despite the exclusion of 20 of the 25 DE genes initially found on this chromosome. Two of the five remaining genes on Chromosome 24 at DAR = 0.3 can be found in The Synteny Database. These genes exist on medaka Chromosomes 17 and 20 indicating their aggregation onto Chromosome 24 during the past ~323 million years, see S7 Fig.

## DAR-based weighting in GSEA

GSEA requires a ranked list of genes to calculate gene set enrichment scores. In typical GSEA, a ranking statistic is calculated from the results of differential expression analysis, which is used to define the order of the ranked list. Often, the ranking statistic is calculated by multiplying the sign of each gene's $\log_2$ fold-change value by $-\log_{10}$ of its respective *p*-value, resulting in the most significant up-regulated genes being positioned at the top of the list, and the most significant down-regulated genes at the bottom. Genes at the top or bottom of the list provide a greater contribution to their associated pathways' enrichment scores than those in the middle of the list. To deemphasise the contribution to differential expression of genes more likely to be eQTLs, we weighted each gene's ranking statistic prior to GSEA by multiplying it by the complement of the gene's DAR value (1 –DAR). See S8 Fig for the impact of weighting on gene rankings in the *naglu*$^{A603Efs/A603Efs}$ vs. *naglu*$^{+/+}$ 7 dpf sibling larval transcriptome dataset. This resulted in genes existing in high DAR regions being pushed towards the middle of the list, where they would have less impact on the GSEA algorithm. Weighting gene ranks using the DAR metric in the *naglu*$^{A603Efs/A603Efs}$ vs. *naglu*$^{+/+}$ dataset caused one pathway that was not initially significant, *Glycosaminoglycan Degradation*, to achieve significance (FDR-adjusted *p*-value = 1.56e-2). Given the expectation of *Glycosaminoglycan Degradation* being a true positive, this result supports DAR-weighting as a viable strategy when using GSEA. The other five initially significant pathways, *Lysosome*, *Focal Adhesion*, *Other Glycan Degradation*, *ECM Receptor Interaction*, and *Spliceosome*, all retained their significance when using the weighted approach (Fig 8).

We also examined the effects of DAR-weighting in our separate brain transcriptome dataset from 6-month-old zebrafish that compares a heterozygous mutation in the *sorl1* gene

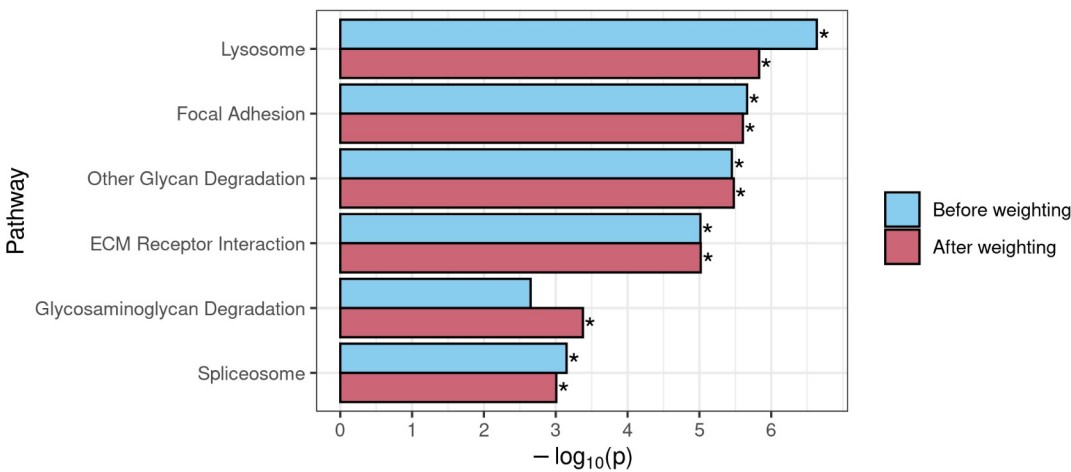

**Fig 8. Comparison of GSEA results for KEGG gene sets that achieved significance before and/or after use of DAR to weight the gene-level ranking statistic in the *naglu*^A603Efs/A603Efs vs. *naglu*^+/+ 7 dpf sibling larval transcriptome dataset. An asterisk indicates that a pathway was determined to be significantly enriched (FDR-adjusted *p*-value < 0.05).**

(*sorl1*^V1482Afs/+, an EOfAD-like model) to wild-type siblings [21]. This dataset displayed CC-DEGs (S1 Fig) and DAR levels of around 0.4 were noted as a concern (S9 and S10 Figs). A comparison of GSEA results before and after weighting also showed compelling results for pathways we expect to be perturbed by the mutation (S11 Fig). Gene sets associated with neurodegeneration, i.e. *Parkinsons disease*, *Huntingtons disease* and *Alzheimers disease*, all showed increased significance using the DAR-weighting method. Additionally, *Oxidative phosphorylation*, the most significantly enriched gene set and one we see consistently enriched in all our zebrafish models of EOfAD, also increased in significance. Three of the pathways (*Viral myocarditis*, *Cytokine-Cytokine Receptor Interaction* and *Galactose metabolism*) that were initially classified as significant, but were on the verge of non-significance, were shown to lose significance after DAR-weighting.

## DAR-based weighting in over-representation analysis

Over-representation analysis differs from the approaches of ROAST and GSEA by determining if a higher-than-expected representation of DE genes exist within chosen gene sets (e.g. KEGG pathways). We sought to broaden our investigation into the utilisation of DAR with existing enrichment methods by performing over-representation analysis on Gene Ontology (GO) terms. A standard approach in over-representation analysis is to perform a hypergeometric test. However, in RNA-seq data, selection bias attributed to transcript length [40] violates the assumptions behind a hypergeometric distribution. The *GOseq* method [41] addresses the implications of selection bias in RNA-seq datasets by initially calculating a Probability Weighting Function (PWF) from data representing the source of bias, such as transcript length. The PWF is then used to weight the chance of selecting each gene when forming a null distribution for gene set membership. However, the bias data used to calculate a PWF need not be restricted only to transcript length. A PWF can be formulated using any type of data that is expected to influence the classification of a gene as being DE.

To test the utility of the DAR metric in a PWF we implemented GO over-representation analysis with the *GOseq* method and compared the outcomes of two different choices of bias data: transcript length and DAR. We chose the MPS IIIB relevant *naglu*^A603Efs/A603Efs vs. *naglu*^+/+ data to illustrate our findings, as this dataset contains a high number of DE genes

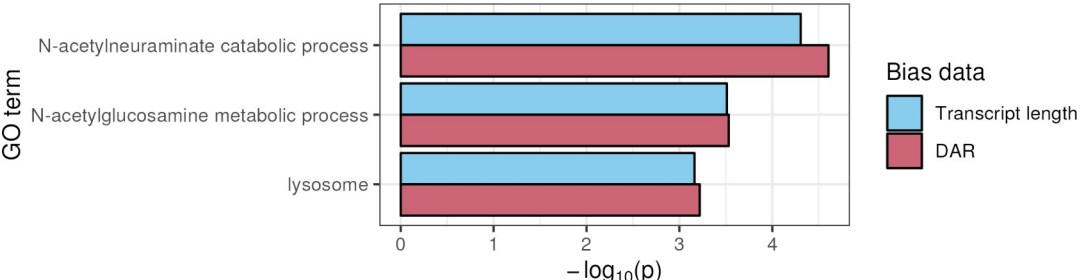

**Fig 9. Comparison of over-representation analysis results for GO terms that achieved significance when using either transcript length or DAR as bias data for *GOseq* analysis of the *naglu*^A603Efs/A603Efs vs. *naglu*^+/+ 7 dpf sibling larval transcriptome dataset.** All three displayed GO terms showed statistical significance when transcript length or DAR was used as bias data. However, DAR showed greater significance for all terms.

relative to other datasets and our knowledge of the biological effects of mutations in the corresponding human gene (*NAGLU*) is quite detailed.

Firstly, we compared plots of the PWF calculated from either transcript length or DAR to quantify the amount of respective bias present in the data (S12 Fig). Minimal transcript length bias was observed, while genes with high DAR showed a clear bias towards classification as DE. This means that transcript length has relatively less influence than DAR on the formation of the null distribution for assessing over-representation in this dataset, and the outcomes can be interpreted as being more similar to that of a typical hypergeometric test. We then identified those GO terms showing significant over-representation of DE genes (Fig 9). Both choices of bias data returned the same three GO terms as significant (FDR-adjusted $p$-value $< 0.05$). These GO terms are known to be implicated in MPS IIIB pathology: *N-acetylneuraminate catabolic process*, *N-acetylglucosamine metabolic process* and *lysosome*. However, similar to the outcomes of using DAR to weight the GSEA ranked list, gene sets implicated in MPS IIIB pathology showed greater statistical significance when DAR was used as bias data, suggesting that DAR may be suitable for the improvement of over-representation analysis.

## Discussion

High throughput RNA-sequencing is currently the most advanced technology for gene expression analysis. However, experimental artifacts are widespread and can lead to misinterpretation of results when not appropriately addressed. There is considerable evidence for transcript length and guanine-cytosine (GC) content as complexities of RNA-seq contributing to bias in differential expression and functional gene set analyses [42,43]. Techniques such as *cqn* [44], *EDASeq* [45] and *GOSeq* [41] have been developed to assist in the correction of these issues. Biological artefacts resulting from less-understood mechanisms such as transcriptional bursting (reviewed in [46]) also pose questions about the reliability of inferences from RNA-seq data. As we uncover new challenges in the interpretation of RNA-seq data, we require the continuous development of procedures to identify and alleviate such situations.

White et al. [10] recently proposed that CC-DEGs might be mistakenly interpreted as differential expression caused by a mutation of interest when, in reality, they are an allele-specific expression (eQTL)-dependent effect. The authors highlighted that eQTLs can be revealed in contexts of differential allelic representation between sample groups subject to comparison; a situation commonly encountered when using non-isogenic strains of model organisms. White et al. used inbred descendants of a cross between two discrete isogenic wild-type zebrafish strains and identified chromosomal sub-regions of homozygosity corresponding to one or other of the original strains. When gene expression data from these individuals was grouped

according to such regional homozygosity, and then compared according to its derivation from the original strains, evidence was revealed for an eQTL basis of differential expression. While the analysis was appropriately designed to address the eQTL issue, the regions of chromosomal homozygosity analysed were relatively small and lacked the genetic context that would commonly be encountered in RNA-seq experiments investigating the effects of mutations. Generally, homozygosity for a mutant or wild-type sequence is the criterion for sorting individual samples into comparison groups, resulting in samples homozygous for a large portion of a mutation-bearing chromosome near the site of the mutation. The majority of the 82 homozygosity regions analysed by White et al. were limited to 3Mbp or less, which accounts for ~6% of the average length of zebrafish chromosomes (GRCz11, Ensembl version 101). In the context of a zebrafish transcriptome comparison such as described in Fig 1, recombination will often alter the size of the shared homozygous region to a length shorter than an entire chromosome. However, the region of homozygosity around the mutation locus may remain larger than 3 Mbp, particularly as recombination is suppressed during meiosis in male zebrafish [29].

Our discovery of clustering of, apparently, non-eQTL DE genes around heterozygous *psen1* mutations is consistent with Fisher's assertion that functional relationships between genes create selective pressure driving the chromosomal co-location of advantageous allelic pairs during evolution to allow linkage disequilibrium. However, the differential representation of eQTLs in groups under comparison is also a parsimonious explanation for CC-DEGs. In an attempt to resolve these two explanations, we developed a means to measure localised between-group differential allelic representation in RNA-seq datasets. Our DAR metric allows for refined identification of genomic regions that may be prone to eQTL expression bias in transcriptome studies. Such bias was clearly highlighted in our analysis of the mouse *APOE* dataset (Fig 4) where we found extreme clustering of genes around the *APOE* locus, apparently differentially expressed due to differential representation of eQTLs in the comparison groups. By the same measure, the CC-DEGs associated with heterozygous, dominant *psen1* mutations are unlikely to be eQTLs. Intriguingly, the CC-DEGs differ for the functionally very distinct *psen1*$^{T428del}$ EOfAD-like and *psen1*$^{W233fs}$ fAI-like mutations, and the degree of clustering of the CC-DEGs is consistent with the selective disadvantage that each mutation would cause. In particular, the fAI-like mutation, which we have shown to affect significantly Notch signalling [12] (a mechanism of central importance to cell differentiation and embryo development, reviewed in [47]) shows differentially expressed genes scattered across the entire length of Chromosome 17, in contrast to the fewer, tightly clustered DE genes of the EOfAD-like mutation that, in humans, causes an adult-onset disease. In male zebrafish, the entirety of Chromosome 17 shows a recombination distance of less than 35 cM (~118 cM in females) [29] meaning that alleles of any pair of genes on this chromosome can exist in a state of linkage disequilibrium and so might be under positive selection for chromosomal co-location if co-involved in Notch signalling-dependent processes.

Our limited investigation of chromosome evolution in zebrafish revealed some evidence consistent with the idea that selection for linkage disequilibrium may be influential during genome evolution. A number of CC-DEGs due to mutation of *psen1* were found to reside on separate chromosomes in an ancestral chromosome arrangement. However, a far more expansive and detailed investigation of this issue would be necessary for confidence that generation of linkage disequilibrium is an important positive selective pressure on chromosomal rearrangements. If such selective pressure is important, then why have CC-DEGs not been observed more widely in transcriptome studies? It is notable that the only situations where our laboratory has observed distinct CC-DEGs patterns are in well-controlled comparisons of single gene mutations in 2 dpf embryos [8], 7 dpf larvae, or in young (~6-month-old zebrafish & ~3-month-old mouse) whole brains/cortices. If genes are connected in complex, homeostatic

networks, then disturbance of only one network node would cause great expression changes in only the most closely co-functional nodes, much as pulling on one point in a spider's web causes greatest displacement for the most closely associated threads. In contrast, influences that disturb multiple nodes simultaneously (such as environmental change) would, like wind over a spider's web, give much more widespread network expression change. The magnitude of a mutation's phenotypic effect may also be a factor. In our analysis of human *APOE* alleles in humanised mouse cortices, the *APOE4* allele may differ in its activity from *APOE3* more significantly than the *APOE2* allele differs from *APOE3*, contributing to the greater number and broader chromosomal distribution of genes that are DE in the *APOE4/4 vs. APOE3/3* comparison than the *APOE2/2* vs. *APOE3/3* comparison.

Identification of possible eQTLs in transcriptome comparisons is important if their exclusion from analyses can improve the identification of affected gene sets. Our investigation into utilising the DAR metric to exclude possible eQTLs in the *naglu*$^{A603Efs/A603Efs}$ vs. *naglu*$^{+/+}$ 7 dpf sibling larval transcriptome dataset revealed a number of interesting observations. Both ROAST and GSEA methods returned the expected *Glycosaminoglycan Degradation* gene set as significantly enriched at various stages of the thresholding procedure. ROAST, however, appeared far more robust to the removal of genes in comparison to GSEA. ROAST consistently reported *Glycosaminoglycan Degradation* as significantly enriched at most DAR thresholds, together with two other gene sets known to be implicated with MPS IIIB disease (*Other Glycan Degradation*, and *Lysosome*). These gene sets also remained consistent in their respective rankings as defined by their *p*-values. GSEA, on the other hand, displayed considerably more variability in its outcomes. Aside from the four gene sets that remained significant at all thresholds using GSEA (*Lysosome*, *Focal Adhesion*, *Other Glycan Degradation*, *ECM Receptor Interaction*), we noticed that excluding genes had a clear impact on the relative rankings of gene sets. We also observed a general increase in the number of significant gene sets as more genes from high DAR regions were excluded, suggesting that the removal of eQTL-biased genes may improve the sensitivity of GSEA. This is indeed evident for the *Glycosaminoglycan Degradation* pathway, which only achieved significance at a DAR threshold of 0.6. It is likely that the *Glycosaminoglycan Degradation* pathway/gene set did not achieve significance at a threshold greater than 0.6 because the genes of this dataset mostly have DAR values below this threshold. Therefore, removal of biased genes with DAR greater than 0.6 allowed for those genes associated with *Glycosaminoglycan Degradation* to increase their relative ranking among the genes remaining. Similarly, at a DAR below 0.6, sufficient genes of the *Glycosaminoglycan Degradation* gene set were removed for this gene set to lose significance. The detection of the MPS IIIB-related *Glycosaminoglycan Degradation* pathway in this data set only at a DAR threshold of 0.6 suggests that this DAR threshold is the most suitable for removing bias from this particular dataset. However, the optimal DAR threshold is likely to vary between datasets so that its choice becomes arbitrary. Therefore, we considered alternative solutions to this issue, i.e. weighting-based approaches.

The disparity in the effects of thresholding between ROAST and GSEA is considerable but not surprising. Indeed, GSEA is well-known to be prone to bias, such as the false positive calls attributed to sample-specific gene length bias by Mandelboum et al. [42]. GSEA requires a gene-level ranked list as determined by differential expression analysis to calculate downstream gene set scores. It is also a competitive test, meaning that the magnitude of enrichment of each gene set affects the significance assessment of other gene sets. Therefore, biased genes influence the ranking of genes showing mutation-driven differential expression, and gene sets containing the biased genes affect the significance of gene sets without them. In comparison, ROAST is a non-competitive test that works directly from the gene expression values, without requiring intermediate calculations of gene-level scores. Therefore, in the context of ROAST,

biased genes will only impact the outcomes for a particular gene set if they are a member of that gene set. This provides a likely explanation for ROAST appearing more robust to experimental biases than GSEA.

In terms of good bioinformatics practise, excluding information is generally undesirable in contrast to applying statistical methods such as gene-level weightings to address differences in confidence. The DAR-based weighting approach for GSEA presented compelling findings because *Glycosaminoglycan Degradation*, a pathway we expect to see enriched in studies of MPS IIIB, only achieved significance after the ranked list was weighted with DAR, along with all other gene sets, which showed consistent significance across all thresholds. While the gene sets we expected to see enriched in the $sorl1^{V1482Afs/+}$ vs. $sorl1^{+/+}$ comparison were already statistically significant before weighting, the increase in significance after weighting further reinforces the validity of this method. This finding provides confidence that using DAR to weight ranking statistics and adjust gene rankings can be a suitable approach to improving GSEA in situations where eQTLs are present. While this approach is not applicable for ROAST, as it does not require intermediate gene-level rankings, ROAST appears far more robust to eQTL artefacts and may simply be a preferable choice in many situations.

The application of DAR to calculate a PWF for the *GOseq* method showed promising results for the improvement of over-representation analysis. This provides a further example of where DAR can be utilised as a weighting technique for statistical analysis without the removal of data. However, due to the fact that there was minimal length bias in the *naglu* dataset, the question arises as to whether it is more important to account for transcript length or DAR when length bias is, in fact, present. In this situation we suggest that length bias is accounted for prior to differential expression testing through well-established methods such as *cqn* [44], allowing the impacts of DAR to be addressed downstream when performing over-representation analysis.

## Materials and methods

### Data description and generation

Two of the RNA-seq datasets involved in this analysis have already been described in previously published analyses. For information about animal ethics, genome editing, breeding strategy and RNA-seq data generation for each dataset please refer to their respective publications. The *psen1* mutant zebrafish dataset is described in [12] and is available in the GEO database (GSE164466). The *APOE* mouse dataset is described in [26] and is available at the AD Knowledge Portal (accession number syn20808171, https://adknowledgeportal.synapse.org/). The $naglu^{A603Efs/A603Efs}$ vs. $naglu^{+/+}$ 7 dpf sibling larval RNA-seq dataset is available from the GEO database (GSE217196). It is comprised of RNA-seq libraries derived from whole zebrafish 7 dpf sibling larvae either heterozygous for the EOfAD-like mutation $psen^{Q96\_K97del}$, homozygous for the MPSIIIB-like mutation $naglu^{A603Efs}$, or wild-type.

### Differential expression analysis

Paired-end raw RNA-seq reads in FASTQ format were assessed for quality with *FastQC* [48] and *ngsReports* [49]. Adapters, poly-G regions and bases with a PHRED score below 20 were trimmed from raw RNA-seq data using *fastp* (v0.23.2) [50]. Any resulting reads were removed if they were shorter than 35bp after trimming. Trimmed reads were aligned to the Ensembl genome (GRCz11 release 101 for zebrafish, GRCm39 release 104 for mouse) using *STAR* (v2.7.7a) [51] in two-pass mode to achieve better alignment around novel splice junctions. Aligned reads were counted at the gene-level using *featureCounts* from the *Subread* package (v2.0.1) [52] if they were unique and mapped to strictly exonic regions. The analysis of count

data was then performed in R [25]. Genes were retained for downstream analysis if they had a minimum of 1 count per million in at least the number of samples equalling the smallest genotype group. Differences in library sizes was accounted for by normalisation using the trimmed mean of M-values method [53]. Conditional Quantile Normalisation (*CQN*, [44]) was performed prior to differential expression analysis if guanine-cytosine content or length bias was detected in the dataset. Differential expression analysis was performed by fitting a negative binomial generalized log-linear model to gene-level counts and likelihood ratio testing using *edgeR* [54]. The design matrix was specified with an intercept of 0 and coefficients corresponding to genotypes. Contrasts were defined and the null hypothesis was tested that the specified contrasts of the coefficients are equal to zero. Genes were considered to be differentially expressed in each comparison if the false discovery rate (FDR)-adjusted *p*-value was less than 0.05.

## Variant calling and the DAR metric

Variants were determined based on the Genome Analysis Toolkit (GATK) best practices workflow for RNA-seq short variant discover (SNPs + Indels) [24]. Unique Molecular Identifiers (UMIs) were incorporated into FASTQ headers with *fastp* (v0.20.1) where this information was available to improve deduplication methods. Adapters were trimmed from raw reads using *fastp* (v0.20.1) and resulting trimmed reads were discarded if they were shorter than 35bp. Trimmed reads were aligned to the Ensembl genome (GRCz11 release 101 for zebrafish, GRCm39 release 104 for mouse) using *STAR* (v2.7.7a) in two-pass mode. Read groups were assigned to the aligned reads based on sample and sequencing lane with the GATK's (v4.2) *AddOrReplaceReadGroups* to allow accurate correction of technical batch effects. Reads were grouped based on their UMI and mapping coordinates by tagging BAM files with *umitools group* (v1.1.1) [55]. Reads were deduplicated with Picard *MarkDuplicates* (GATK v4.2) and those that spanned splicing regions were split using *SplitNCigar* reads (GATK v4.2). A set of known variants was obtained from the Ensembl database for the corresponding release chosen for the Ensembl genome during alignment. Base Quality Score Recalibration (BQSR) was performed with *BaseRecalibrator* and *ApplyBQSR* (GATK v4.2) to detect systematic errors in sequencing and adjust base quality scores accordingly. Variants were called in GVCF mode with *HaplotypeCaller* and joint-genotyped with *GenotypeGVCFs* (GATK v4.2). The resulting calls were filtered based on the GATK's recommended specific hard filters (phred-scaled *p*-value using Fisher's exact test for strand bias (FS) > 60, variant confidence/quality by depth (QD) > 2, root mean square mapping quality (MQ) < 40, strand odds ratio (SOR) > 4). Lastly, variants were selected for only single nucleotide polymorphisms (SNPs) using the GATK (v4.2) *SelectVariants* tool.

Calculation of the DAR metric was performed in R. Genotype calls were assessed at each SNP locus that contained a non-reference allele in at least one sample. Samples were grouped according to mutant genotype and SNPs were excluded from the analysis if the number of samples with missing or filtered genotype calls was greater than 50% within each group. Allele-level counts for each SNP were normalised as a proportion of the total number of reported alleles for each group (2n alleles reported for n genotypes called). The Euclidean distance formula was then implemented using the base R *dist* function on the normalised allele counts between the sample groups contrasted during differential expression analysis. The calculated distance was divided by its maximum possible value of $\sqrt{2}$ to achieve the DAR metric ranging between 0 and 1.

## Assigning gene-level DAR values

Because not all genes contained SNP information to assign a DAR value, we smoothed the DAR metric using an elastic window approach of n = 11 SNPs, allowing for a broader

representation of DAR. Genes were assigned a DAR value based on the average of any regions the gene was located in.

## DAR-based thresholding for functional enrichment analysis

Prior to functional enrichment analysis, gene exclusion thresholds were established at DAR values ranging from 0.1 to 1.0 at intervals of 0.1. Here, any genes that had an assigned DAR value above the threshold were excluded from the analysis. Two separate methods were chosen to perform functional gene set analysis: ROAST [33], implemented with the *fry* function from the *limma* package [34], and GSEA [35], implemented with the *fgseaMultilevel* function from the *fgsea* package [36], Both methods were performed at each of the 10 established DAR thresholds.

## DAR-based weighting for functional enrichment analysis

We evaluated GSEA under two conditions: with and without DAR-based weighting. GSEA without DAR weighting involved creating a ranked list, where the order was determined by a ranking statistic that prioritises the most significantly upregulated genes at the top and the most significantly downregulated genes at the bottom of the list. In the weighted approach, the ranking statistic for each gene was adjusted by multiplying it with the complement of the gene's associated DAR value (i.e. 1 –DAR).

Over-representation analysis was performed using the *goseq* function from the *goseq* package [41]. Each gene's median transcript length or DAR value was used to calculate a PWF for weighting the chance of selecting a gene when forming the null distribution for gene set membership. GO terms with no DE genes or less than 5 total genes were excluded from the analysis.

## The Synteny Database

Analysis of conserved synteny between zebrafish and medaka was performed using the dot plot feature of the Synteny Database [32]. The *Danio rerio* Ensembl version 71 genome was selected as the source genome and the *Oryzias latipes* Ensembl version 71 genome was selected as the outgroup genome. Note that Ensembl version 71 was selected as it was the most recent available release on the Synteny Database. Plots were generated by inputting the zebrafish Ensembl gene identifiers, and selecting the corresponding chromosome to display along the x-axis.

## Supporting information

**S1 Fig. Manhattan plots highlighting CC-DEGs from differential expression testing in heterozygous vs. wild-type zebrafish siblings (6-month-old brains) and between homozygous mouse mutants (3-month-old male cortices).** A) zebrafish EOfAD-like $psen1^{T428del/+}$ vs. $psen1^{+/+}$ B) zebrafish EOfAD-like $sorl1^{R122Pfs/+}$ vs. $sorl1^{+/+}$ C) zebrafish EOfAD-like $sorl1$-$^{V1482Afs/+}$ vs. $sorl1^{+/+}$ D) mouse APOE4/4 vs. APOE3/3. Non-random accumulation of DEGs was tested using a Bonferroni-adjusted Fisher's exact test *p*-value for enrichment of DE genes on the mutant chromosome, A) $psen1^{T428del/+}$ vs. $psen1^{+/+}$: $p$ = 8.63e-3, B) $sorl1^{R122Pfs/+}$ vs. $sorl1^{+/+}$: $p$ = 1.92e-4, C) $sorl1^{V1482Afs/+}$ vs. $sorl1^{+/+}$: $p$ = 9.24e-2, D) APOE4/4 vs. APOE3/3: $p$ = 1.00. Genes are plotted along the x-axis based on their chromosomal position in alternating shades of grey for visual distinction between chromosomes. Genes on the chromosome containing the mutation are highlighted in red. The raw *p*-values are plotted along the y-axis at the -$\log_{10}$ scale such that the most significant genes exist at the top of the plot. The cut-off for gene

differential expression (FDR-adjusted $p$-value $< 0.05$) is indicated by a dashed horizontal line. Genes classified as differentially expressed under this criterion are represented as diamonds with a black outline.
(TIFF)

**S2 Fig. Elastic window sizes summarised in 100 bins of equal size for A) the mouse *APOE* dataset B) the zebrafish *psen1* dataset.** Window sizes are determined based on the genomic distance between 11 subsequent SNPs. Zebrafish datasets have a greater number of variant sites distributed across the genome relative to mouse due to the lack of isogenicity, resulting in much smaller window sizes.
(TIFF)

**S3 Fig. The relationship between DE genes and DAR along the entirety of mouse Chromosome 7 between female *APOE2/2* and *APOE3/3* mouse cerebral cortices (at 3 months of age).** The plot contains four sets of information represented by separate tracks horizontally. Track A represents the axis of Chromosome 7. The position of the *APOE* gene is marked and labelled in bold red. Track B displays differentially expressed genes according to their positions along the chromosome. 183 of 1126 total genes (16.25%) on Chromosome 7 that were expressed in the dataset were classified as DE (FDR $< 0.05$). Track C shows the trend in DAR as a connected scatterplot with each point in black representing the DAR value at a single nucleotide variant position (elastic sliding window size = 11 variants). Positions of the DE genes shown in track B are indicated by light red lines.
(TIFF)

**S4 Fig. Cumulative distribution of DAR by chromosome between male *APOE2/2* and *APOE3/3* mouse cortices (3 months).** The mutant chromosome exhibits the most regions of high DAR.
(TIFF)

**S5 Fig. Dotplot displaying the chromosomal location of medaka (Ola) genes predicted by the Synteny Database [32] to be orthologues of zebrafish (Dre) Chromosome 17 DE genes from the fAI-relevant, *psen1*$^{W233fs/+}$ vs. *psen1*$^{+/+}$ comparison.** Genes are plotted along the x-axis based on their position on Chromosome 17 in zebrafish, while the y-axis indicates the chromosome they are located on in medaka. The four DE genes predicted to have orthologues are circled in blue. The two genes without a secondary blue circle plotted along the y-axis (*dglucy* and *psen1*), were predicted by the Synteny Database to exist on alternate scaffolds of medaka Ensembl version 71, which are not plotted. The smaller grey-filled circle indicates the centromere of Dre Chromosome 17.
(TIFF)

**S6 Fig. The effects of gene exclusion by DAR threshold on KEGG gene set *p*-values from GSEA testing.** The gene sets displayed are those that achieved significance for at least one DAR threshold value. $p$-values are plotted on a $-\log_{10}$ scale along the y-axis such that the most significant results exist at the top of each graph. Each position along the x-axis represents a different DAR gene exclusion threshold. Dots on the graph filled with colour indicate that the gene set was classified as significantly enriched (FDR-adjusted $p$-value $< 0.05$). The number inside a dot corresponds to the number of leading-edge genes that contributed to the respective gene set's enrichment score.
(TIFF)

**S7 Fig. Dotplot displaying the chromosomal location of medaka (Ola) genes predicted by the Synteny Database [32] to be orthologues of zebrafish (Dre) Chromosome 24 DE genes**

from the *naglu*$^{A603Efs/A603Efs}$ vs. *naglu*$^{+/+}$ 7 dpf sibling larval RNA-seq dataset comparison at a DAR threshold of 0.3. Genes are plotted along the x-axis based on their position on Chromosome 24 in zebrafish, while the y-axis indicates the chromosome they are located on in medaka. The two DE genes predicted to have orthologues are circled in blue. The smaller grey-filled circle indicates the centromere of Dre Chromosome 24.
(TIFF)

**S8 Fig. The effects of using DAR to weight gene-level rankings in the MPS IIIB-relevant transcriptome comparison of *naglu*$^{A603Efs/A603Efs}$ vs. *naglu*$^{+/+}$ 7 dpf sibling larvae.** Genes that are plotted close to the blue diagonal line are least impacted by the weighting method. Genes that exist on the mutant chromosome (Chromosome 24) are coloured red, while those that exist on other chromosomes are coloured black. Genes that were most substantially affected (rank change > 2000) are labelled with their respective gene symbol.
(TIFF)

**S9 Fig. The relationship between DAR and DE genes along Chromosome 15 in the comparison of *sorl1*$^{V1482Afs/+}$ vs. *sorl1*$^{+/+}$ sibling zebrafish brains.** The plot contains four sets of information represented by separate tracks. Track A represents the axis of Chromosome 15. The position of the *sorl1* gene is marked and labelled in bold red. Track B displays differentially expressed genes according to their positions along the chromosome. Track C shows the trend in DAR as a connected scatterplot with each point representing the DAR value at a single nucleotide variant position (elastic sliding window, n = 11 variants). Positions of the DE genes shown in track B are indicated by light red lines.
(TIFF)

**S10 Fig. Cumulative distribution of DAR by chromosome between *sorl1*$^{V1482Afs/+}$ and *sorl1*$^{+/+}$ sibling zebrafish.** The mutant chromosome exhibits the greatest number of regions of high DAR.
(TIFF)

**S11 Fig. Comparison of GSEA results for KEGG gene sets that achieved significance before and/or after using DAR to weight the gene-level ranking statistic in the *sorl1*$^{V1482Afs/+}$ vs. *sorl1*$^{+/+}$ siblings dataset.** An asterisk is used to denote that the pathway was determined to be significantly enriched.
(TIFF)

**S12 Fig. The resulting fit of the Probability Weighting Function (PWF) on the *naglu*$^{A603Efs}$ dataset for bias data A) Median transcript length B) DAR.** The points indicate the proportion of DE genes for bias data in 200 gene bins. The green line represents the monotonic spline fitted when calculating the PWF. Minimal transcript length bias is observed relative to DAR bias.
(TIFF)

## Acknowledgments

The authors wish to thank Julian Catchen for assistance with the use of the Synteny Database. The results published here are in whole or in part based on data obtained from the AD Knowledge Portal (https://adknowledgeportal.org). Support for these studies was provided by the NIH RF1 AG051504 and P01 AG030128. We thank Drs. Patrick Sullivan and Nobuyo Maeda for generating human APOE targeted replacement mice and providing access through Taconic. The zebrafish transcriptome data was produced with assistance from the South

Australian Genomics Centre. This work was also supported with supercomputing resources provided by the Phoenix HPC service at the University of Adelaide.

## Author Contributions

**Conceptualization:** Lachlan Baer, Karissa Barthelson, David L. Adelson, Stephen M. Pederson, Michael Lardelli.

**Data curation:** Karissa Barthelson.

**Formal analysis:** Lachlan Baer, Karissa Barthelson.

**Investigation:** Lachlan Baer, Karissa Barthelson.

**Methodology:** Lachlan Baer, Karissa Barthelson, John H. Postlethwait, Stephen M. Pederson.

**Project administration:** Lachlan Baer, Stephen M. Pederson, Michael Lardelli.

**Software:** John H. Postlethwait.

**Supervision:** Stephen M. Pederson, Michael Lardelli.

**Validation:** Lachlan Baer.

**Writing – original draft:** Lachlan Baer, Michael Lardelli.

**Writing – review & editing:** Lachlan Baer, Karissa Barthelson, John H. Postlethwait, David L. Adelson, Stephen M. Pederson, Michael Lardelli.

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
