## [Decision Letter · Decision Letter 0]

10 Sep 2023

Dear %TITLE% Lardelli,

Thank you very much for submitting your manuscript "Differential allelic representation (DAR) identifies candidate eQTLs and improves transcriptome analysis" for consideration at PLOS Computational Biology.

As with all papers reviewed by the journal, your manuscript was reviewed by members of the editorial board and by several independent reviewers. In light of the reviews (below this email), we would like to invite the resubmission of a significantly-revised version that takes into account the reviewers' comments.

We cannot make any decision about publication until we have seen the revised manuscript and your response to the reviewers' comments. Your revised manuscript is also likely to be sent to reviewers for further evaluation.

Sincerely,

Alexandre V. Morozov, Ph.D.

Academic Editor

PLOS Computational Biology

William Noble

Section Editor

PLOS Computational Biology

Reviewer's Responses to Questions

**Comments to the Authors:**

Reviewer #1: This paper investigates the phenomenon of functionally related genes becoming linked over evolution. The authors analyse several different mutants in both mouse and zebrafish which show an excess of differentially expressed (DE) genes on the same chromosome as the mutation. They propose a statistic (DAR) to distinguish between genes that are truly DE as a result of the mutation and those that appear to be DE due to the segregation of linked eQTLs into the two different sample groups. They provide a SnakeMake workflow to process RNA-seq including calculating DAR. Using DAR, the authors show, in a comparison of two different human APOE alleles introduced into mice, that the linked DE genes are more likely to be caused by eQTLs. In contrast, two psen1 mutations in zebrafish, show low DAR values across the whole of the chromosome with the mutation. This suggests that eQTLs are likely not the cause of these DE genes. Further, the authors suggest that these genes have become linked over the course of evolution. They look at Medaka as an example. Four of the seven DE genes in the psen1(W233fs) comparison (including psen1 itself) are located on the same Medaka chromosome. Of the other three, two had identifiable Medaka orthologues that are located on different Medaka chromosomes. The authors suggest that these two genes (fam167aa and sh3pxd2ab) have been captured at some point since the divergence of Medaka and zebrafish.

To investigate the effect of distinguishing true DE genes from those caused by eQTLs, the authors look at the effect on the gene set enrichment methods, ROAST and GSEA. They do this in another zebrafish example (naglu-A603Efs) by using a thresholding approach to progressively remove genes with high DAR scores that are more likely to represent DE genes caused by eQTLs. ROAST, possibly due to its methodology seems to be quite robust to the inclusion of eQTL-driven DE genes, whereas terms enriched using GSEA fluctuate more as the threshold changes. The authors propose that for GSEA, rather than using the DAR score to remove genes from the gene list, it can be used to downweight suspect DE genes in the input. Using this DAR-weighting strategy, one of the gene sets expected to be enriched (Glycosaminoglycan Degradation) becomes significantly enriched when previously it wasn't.

More and more comparative transcriptomic datasets are being generated from samples not from inbred lines. Therefore, these issues are likely to become more widespread. The analysis is well done, and all the relevant code is available. The Rmarkdown produced html files make browsing the results really easy.

Major points:

It would be very helpful to have a table with the analysed experiments and number of total and mutation-linked DE genes in each.

The analysed DE lists seem very short (apart from the APO mouse data), so it would be good to see a couple of examples where their method is applied to DE lists longer than 50. For example, White et al. has a few suitable RNA-seq experiments.

The authors look at two methods of Gene Set enrichment testing (ROAST and GSEA). Frequently use tools such as PANTHER, gProfiler, Ontologizer and topGO use different enrichment statistics. It would be really useful to see what effect DAR-based thresholding has on one of these methods as a comparison.

I can't reconcile the data in the 211130_Q96K97del_A603fs repository (211130_Q96K97del_A603fs-master/docs/enrichment.html) and the -log10(p) values plotted in Figure 8. The text states that initially five pathways were significant, but the table shows 10. Am I looking at the wrong table?

Also, for Supplementary Figure 10, the text states

"Three of the pathways (Viral myocarditis, Cytokine-Cytokine Receptor Interaction and Galactose metabolism) that were initially classified as significant, but were on the verge of non-significance, were shown to lose significance after DAR-weighting."

In the Figure, 'Cytokine-Cytokine Receptor Interaction' has an asterisk both before and after weighting and I can't locate 'Galactose metabolism' on the plot.

The text says that this shows the data for sorl1R122Pfs/+ vs wt, but the terms look like they represent the V1482Afs_het results (210216_sorl1_snv-master/docs/enrichment.html).

Minor points:

It would be useful context to also state in the text for each comparison how many DE genes there were in total.

For the APOE2/3 comparison only the data from male mice is shown. Are the same CC-DEGS seen in the comparison in female samples?

The distribution of window sizes is show for the mouse data. Are the sizes in zebrafish comparable?

For the DAR weighting method, the ranking of genes that change the most is shown (Supplementary Figure 7). It would be interesting to also see the effect on the DE genes present on Chr 24.

In Supplementary Figure 4 and 6, what do the smaller grey-filled circle represent?

Reviewer #2: Baer et al. develops a method to analyze transcriptomic datasets in order to shed light on the effect of mutation. In particular, they highlight issues with the existing methods and proposes their method as an alternative. I must admit that the paper could have been written in a better way. Even the abstract was very difficult to understand. While the introduction was much better, some of the other sections were obscure. It would be great if the authors rewrite the various sections for clarity. The question the authors are asking is important. The analysis seems solid. The comparisons, exclusion of false eQTLs, along with the identification of CC-DEGs not necessarily eQTLs, are commendable. Please find my comments is below.

1) Gene expression is bursty for a large number of genes. While the mechanism for bursting is not fully understood, it can give rise to bimodality in gene expression. Without accounting for such variability in gene expression will lead to incorrect conclusions while carrying out inferences. It is important that the authors address this issue.

2) In ‘Using a DAR metric to exclude probable eQTLs improves gene set enrichment analyses‘ section, it's remarkable that the Glycosaminoglycan Degradation pathway does not attain statistical significance at any DAR threshold higher than 0.6. This could mean something physiologically relevant about this pathway that the authors should explore more.

3) In the introduction, the authors write, “Positive selection for the pair of alleles can eventually drive them to fixation in the population – they become the only extant alleles for those two genes”. The ‘only extant alleles’ is not clear to me. What about the possibility of new mutations introducing genetic diversity to the population, resulting in emergence of new alleles? If a new allele pair arises that provides an even greater fitness advantage?

**Have the authors made all data and (if applicable) computational code underlying the findings in their manuscript fully available?**

Reviewer #1: Yes

Reviewer #2: None

PLOS authors have the option to publish the peer review history of their article (what does this mean?). If published, this will include your full peer review and any attached files.

Reviewer #1: No

Reviewer #2: No
---

## [Decision Letter · Decision Letter 1]

29 Jan 2024

Dear %TITLE% Lardelli,

We are pleased to inform you that your manuscript 'Differential allelic representation (DAR) identifies candidate eQTLs and improves transcriptome analysis' has been provisionally accepted for publication in PLOS Computational Biology.

Best regards,

Alexandre V. Morozov, Ph.D.

Academic Editor

PLOS Computational Biology

William Noble

Section Editor

PLOS Computational Biology

Reviewer's Responses to Questions

**Comments to the Authors:**

Reviewer #1: The authors have addressed all of the comments apart from analysing additional experiments with longer gene lists. Unfortunately, the suggested experiments from White et al. are not suitable as they are pooled samples, not individuals.

Table 1 provides a good overview of the datasets. Using DAR as the PWF for GOseq is an interesting idea and extends the analysis to hypergeometric tests.

Also, the changes to the Supplemental figures help to clarify the story and the discrepancies between the text and the figures have been reconciled.

Overall, I am satisfied that the manuscript is suitable for publication.

**Have the authors made all data and (if applicable) computational code underlying the findings in their manuscript fully available?**

Reviewer #1: Yes

PLOS authors have the option to publish the peer review history of their article (what does this mean?). If published, this will include your full peer review and any attached files.

Reviewer #1: No

---

## [Editor Report · Acceptance letter]

7 Feb 2024

PCOMPBIOL-D-23-00494R1 

Differential allelic representation (DAR) identifies candidate eQTLs and improves transcriptome analysis

Dear Dr Lardelli,

I am pleased to inform you that your manuscript has been formally accepted for publication in PLOS Computational Biology. Your manuscript is now with our production department and you will be notified of the publication date in due course.

With kind regards,

Anita Estes
